# *Bifidobacterium longum* subsp. *infantis* CECT 7210 Reduces Inflammatory Cytokine Secretion in Caco-2 Cells Cultured in the Presence of *Escherichia coli* CECT 515

**DOI:** 10.3390/ijms231810813

**Published:** 2022-09-16

**Authors:** Ana I. Álvarez-Mercado, Julio Plaza-Díaz, M. Cristina de Almagro, Ángel Gil, José Antonio Moreno-Muñoz, Luis Fontana

**Affiliations:** 1Department of Biochemistry and Molecular Biology II, School of Pharmacy, University of Granada, 18071 Granada, Spain; 2Institute of Nutrition and Food Technology “José Mataix”, Biomedical Research Center, University of Granada, 18016 Armilla, Spain; 3Instituto de Investigación Biosanitaria ibs.GRANADA, Complejo Hospitalario Universitario de Granada, 18014 Granada, Spain; 4Children’s Hospital of Eastern Ontario Research Institute, Ottawa, ON K1H 8L1, Canada; 5Laboratorios Ordesa S.L., Parc Científic de Barcelona, 08028 Barcelona, Spain; 6Instituto de Salud Carlos III, CIBER Fisiopatología Obesidad y Nutrición (CIBERobn), 28029 Madrid, Spain

**Keywords:** *B. infantis* IM-1^®^, *Bifidobacterium longum* subsp. *infantis* CECT 7210, Caco-2 cells, co-cultures, cytokines, dendritic cells, *Escherichia coli*, probiotics

## Abstract

Previous works have described the activity of *Bifidobacterium longum* subsp. *infantis* CECT 7210 (also commercially named *B. infantis* IM-1^®^) against rotavirus in mice and intestinal pathogens in piglets, as well as its diarrhea-reducing effect on healthy term infants. In the present work, we focused on the intestinal immunomodulatory effects of *B. infantis* IM-1^®^ and for this purpose we used the epithelial cell line isolated from colorectal adenocarcinoma Caco-2 and a co-culture system of human dendritic cells (DCs) from peripheral blood together with Caco-2 cells. Single Caco-2 cultures and Caco-2: DC co-cultures were incubated with *B. infantis* IM-1^®^ or its supernatant either in the presence or absence of *Escherichia coli* CECT 515. The *B. infantis* IM-1^®^ supernatant exerted a protective effect against the cytotoxicity caused by *Escherichia coli* CECT 515 on single cultures of Caco-2 cells as viability reached the values of untreated cells. *B. infantis* IM-1^®^ and its supernatant also decreased the secretion of pro-inflammatory cytokines by Caco-2 cells and the co-cultures incubated in the presence of *E. coli* CECT 515, with the response being more modest in the latter, which suggests that DCs modulate the activity of Caco-2 cells. Overall, the results obtained point to the immunomodulatory activity of this probiotic strain, which might underlie its previously reported beneficial effects.

## 1. Introduction

Modulation of the gut microbiome through the administration of beneficial microbes is nowadays an active area of investigation. Probiotics have been described as exerting health-promoting properties, including, among others, the maintenance of the gut barrier function and the local and systemic modulation of the host immune system [1]. Clinical studies have demonstrated the potential of probiotics against many diseases, such as allergic pathologies (including asthma, rhinitis and eczema), diarrhea, inflammatory bowel disease and viral infection [2,3,4,5,6]. However, generalizations concerning the potential health benefits of probiotics should not be made because their effects tend to be strain-specific [7,8,9]. Several important mechanisms underlying the beneficial effects of probiotics include: the colonization and normalization of perturbed intestinal microbial communities in both children and adults; the competitive exclusion of pathogens and bacteriocin production; the modulation of enzymatic activities related to the metabolization of a number of carcinogens and other toxic substances; and the production of volatile fatty acids (short- and branched-chain fatty acids), which play a role in the maintenance of energy homeostasis and regulation of functionality in peripheral tissues [1]. In addition, probiotics increase the intestinal cell adhesion and mucin production, and modulate the activity of gut-associated lymphoid tissue and the immune system [1,10]. Similarly, probiotic metabolites are able to interact with the brain –gut axis and to play a role in behavior [11]. Most of these mechanisms involve the host’s gene expression regulation in specific tissues, particularly in the intestine and liver. The expression of mucin genes by intestinal epithelial cells can be induced by probiotics. Likewise, the toll-like receptor (TLR) and nucleotide-binding oligomerization domain (NOD)-receptor genes, as well as pro-inflammatory transcription factors, cytokines and apoptosis-related genes, can also be affected by commensal bacteria [12].

Previous works have described the characterization of a probiotic strain isolated from the feces of breastfed infants, identified as *Bifidobacterium longum* subsp. *infantis* and deposited in the Spanish Type Culture Collection under the accession number CECT 7210 [13,14]. This strain has the main properties required of a probiotic, such as resistance to gastrointestinal juices, biliary salts, sodium chloride and low pH, as well as adhesion to intestinal mucus and sensitivity to antibiotics [13].

*B. longum* subsp. *infantis* CECT 7210, also referred to as *B. infantis* IM-1^®^, has been shown to inhibit rotavirus replication in vitro; to protect both cultured cells and BALB/c mice from rotavirus infection; to displace some pathogens, especially *Cronobacter sakazakii* and *Salmonella enterica*; and to prevent the adhesion of *Cr. sakazakii* and *Shigella sonnei* to HT29 cells [13,15]. The antirotaviral activity of *B. infantis* IM-1^®^ seems to be mediated by a small peptide of 11 amino acids produced by an extracellular protease activity by the probiotic [16]. Multiple protective effects of this bacterial strain have been described in piglets, alone and in combination with prebiotics or other probiotics [17,18,19,20]. In a double-blind, randomized, multicenter, placebo-controlled clinical trial, *B. infantis* IM-1^®^ reduced the number of diarrhea episodes and was associated with lower constipation prevalence in healthy term newborns [21].

In this work, we investigated the intestinal immunomodulatory effects of *B. infantis* IM-1^®^ by using the intestinal epithelial cell (IEC) line Caco-2. However, because single cultures do not reflect the interactions occurring in the intestinal mucosa, we also used co-cultures of DCs and IECs, both of which are crucial for intestinal homeostasis. In this co-culture, DCs have been shown to open the tight junctions between IECs, send dendrites outside the epithelium and directly sample bacteria [22]. This system has been validated to study the effects of other probiotic strains [23].

## 2. Results

### 2.1. B. infantis IM-1^®^ Supernatant Protects Caco-2 Cells against Escherichia coli CECT 515

We began by fine-tuning the culture conditions, including both the incubation time and the ratio of Caco-2: *E. coli* CECT 515 cells (Appendix A). The viability of the Caco-2 cells decreased after 4 h of treatment with LPS or *E. coli*, although differences were not statistically significant. After 8 h of incubation, cell viability significantly decreased with either of the tested treatments (Appendix A). Therefore, all of the subsequent experiments with Caco-2 cells were performed with 8-h incubations. In a second step, we investigated the most suitable Caco-2: *B. infantis* IM-1^®^ ratio (Appendix A). The ratios 1:3 and 1:30 did not affect the viability of the Caco-2 cells but the ratio 1:100 was detrimental (Appendix A). To ensure Caco-2 stimulation, subsequent experiments were performed with the 1:30 ratio. Whereas the incubation of Caco-2 cells with LPS or *E. coli* CECT 515 for 8 h resulted in a decrease in cell viability, the presence of *B. infantis* IM-1^®^ did not affect cell viability (Figure 1). A clear protective effect was observed in the Caco-2 cells treated with *B. infantis* IM-1^®^ supernatant in the presence of *E. coli* CECT 515, given that viability reached the values of untreated cells (controls) (Figure 1).

### 2.2. B. infantis IM-1^®^ and Its Supernatant Decrease Secretion of Pro-Inflammatory Cytokines by Caco-2 Cells

In general, the Caco-2 cells treated with LPS and *E. coli* CECT 515 for 8 h secreted higher levels of pro-inflammatory cytokines to the culture medium compared with untreated cells (Figure 2). IL-1β and IL-12 levels were not affected by *B. infantis* IM-1^®^ either in the presence or absence of *E. coli* CECT 515 (Figure 2A,C). However, *B. infantis* IM-1^®^ supernatant reduced IL-12 levels in the presence of *E. coli* in comparison with LPS-treated Caco-2 cells (Figure 2C). Interestingly, *B. infantis* IM-1^®^ and its supernatant were capable of reducing both IFN-γ and IL-6 in the presence of *E. coli* CECT 515 (Figure 2B,D). Additionally, *B. infantis* IM-1^®^ supernatant reduced IL-8 levels both in the presence of *E. coli* (Figure 2E), and *B. infantis* IM-1^®^ decreased the secretion of TNF-α in *E. coli*-treated Caco-2 cells (Figure 2F). Finally, *B. infantis* IM-1^®^ supernatant reduced the secretion of IL-4 in Caco-2 cells incubated in the absence of *E. coli* CECT 515 (Figure 2H).

We also measured the TGF-β levels in the culture medium of Caco-2 cells (Figure 3). All treatments decreased the secretion of TGF-β1 in comparison with LPS-treated cells (Figure 3A), and the secretion of TGF-β2 compared with control cells (Figure 3B). Neither the probiotic nor its supernatant induced changes on any isoform of TGF-β compared with the *E. coli*-treated cells.

Next, we studied whether *B. infantis* IM-1^®^ or its supernatant exerted effects on the expression of three genes related to intestinal immunity and integrity: IL-6, TLR-4 and occludin (Figure 4). The results were consistent, as culturing the Caco-2 cells with *B. infantis* IM-1^®^ together with *E. coli* had a synergistic effect on the expression of the three genes compared to cells treated with *E. coli* alone.

### 2.3. B. infantis IM-1^®^ and Its Supernatant Reduce the Pro-Inflammatory Response by Co-Cultures of Caco-2 and DCs

Unlike the single cultures of Caco-2 cells, in co-cultures of Caco-2 plus DCs, the strongest pro-inflammatory effects in cytokine secretion were observed with LPS and not with *E. coli* CECT 515 (Figure 5). Treatment of the co-cultures with *B. infantis* IM-1^®^ or its supernatant together with *E. coli* CECT 515 resulted in reductions in IL-1β and IFN-γ, compared with the LPS-treated co-cultures (Figure 5A,B). Comparisons with *E. coli* CECT 515-treated co-cultures did not reach statistical significance. Similar trends were observed for IL-6, IL-8 and TNF-α (Figure 5D–F). No changes whatsoever occurred in TGF-β levels (Figure 6).

## 3. Discussion

Previous works have described the activity of *Bifidobacterium longum* subsp. *infantis* CECT 7210 (also named *B. infantis* IM-1^®^) against rotavirus in mice [13,16] and intestinal pathogens in piglets [17,18,19,20], as well as its diarrhea-reducing effect on healthy term infants [21]. In the present work, we focused on the intestinal immunomodulatory effects of *B. infantis* IM-1^®^ and for this purpose we used the epithelial cell line isolated from colorectal adenocarcinoma Caco-2. However, because single cultures do not reflect the interactions occurring in the intestinal mucosa, it is important to develop in vitro systems that include both DCs and epithelial cells, both of which are crucial for intestinal homeostasis. For this reason, we also used co-cultures of human dendritic cells (DCs) from peripheral blood, together with Caco-2 cells. This co-culture is an approach to mimic the in vivo conditions under which DCs can open tight junctions between adjacent epithelial cells and take up bacteria directly from the intestinal lumen [22].

Single cultures of Caco-2 cells, challenged with an enteropathogenic strain of *E. coli*, responded with a higher release of pro-inflammatory cytokines to the medium compared with untreated cells. One of the main findings of this study was the attenuation in the secretion of pro-inflammatory cytokines by Caco-2 cells treated with *B. infantis* IM-1^®^ or its supernatant, in particular that of IL-6, IL-8, IFN-γ and TNF-α. Muñoz-Quezada et al. have described the same effect, particularly a decrease in IL-8 and TNF-α secretion, in Caco-2 cells challenged with *Salmonella typhi* and treated with the probiotic strain *Lactobacillus paracasei* CNCM I-4034 [24]. Interestingly, treatments with *B. infantis* IM-1^®^ or its supernatant also resulted in a reduction in pro-inflammatory cytokine secretion in the co-cultures. Similar results have been described for *L. paracasei* CNCM I-4034 by Bermúdez-Brito et al. in co-cultures of Caco-2 cells and DCs challenged with *S. typhi* [23].

As for anti-inflammatory cytokines, we focused on IL-10 and IL-4. *B. infantis* IM-1^®^ did not affect their secretion, but the results were consistent in single cultures of Caco-2 cells and co-cultures. Our IL-4 and IL-10 results coincide with those obtained by different authors that used other probiotic strains [25,26]. However, production of other anti-inflammatory cytokines by *B. infantis* IM-1^®^ should not be ruled out.

Although we did not investigate the effects of *B. infantis* IM-1^®^ or its supernatant in single cultures of DCs, Bermúdez-Brito et al. have described that the exposure of single cultures of human DCs to *S. typhi* also led to a reduced production of pro-inflammatory cytokines and chemokines in the presence of *L. paracasei* CNCM I-4034 or its supernatant [27]. Under our experimental conditions, the anti-inflammatory effect observed in the co-cultures was somewhat more modest in comparison with the single cultures of Caco-2 cells, which suggests the existence of a bidirectional regulation of cell activity between both cell types.

Another interesting finding of this study was the protective effect that the *B. infantis* IM-1^®^ supernatant exerted on Caco-2 cell viability. Similar effects have been described by other authors using different probiotic strains. For instance, Jayashree et al. reported an increase in the viability of Caco-2 cells challenged with methicillin-resistant *Staphylococcus aureus* upon treatment with *Lactobacillus fermentum* 8711 [28]. A possible explanation for the protective effect of *B. infantis* IM-1^®^ supernatant could be related to the capacity of *B. infantis* IM-1^®^ described by Ruiz et al. 2020 [15] to inhibit the adhesion of *E. coli* to HT-29 and MA-104 cells, and also with the capacity of IM-1^®^ to remove adhered *E. coli* from Caco-2 cells, as described by Jayashree et al. for *S. aureus* and *L. fermentum* 8711 [28]. This displacement effect may be due to phenomena such as microbial antagonism by the antimicrobial substances produced by *L. fermentum* 8711 [28]. In fact, the supernatant of the probiotic strain *Bacillus subtilis* subsp. *spizizenii* ATCC 6633 has been shown to improve the survival of Caco-2 cells challenged with *Clostridium perfringens* type A (NCTC 8239), and its effects are mediated by the bacteriocin subtilisin A [29].

*B. infantis* IM-1^®^ is known to secrete short chain fatty acids (SCFA) to the culture medium. Studies have shown a higher production of butyric, lactic, acetic and valeric acids in the intestines of animals fed with this probiotic strain [17,18,19,20,30,31]. Although we did not analyze the composition of the cell-free supernatant, we speculate that the anti-inflammatory effects observed in this study were due to its SCFA content. Other components of the supernatant might also be responsible for, or contribute to, these effects.

Gene expression analysis revealed upregulations in the IL-6, TLR4 and occludin mRNA levels when co-cultures were incubated with *B. infantis* IM-1^®^ in the presence of *E. coli*. Upregulation of occludin points to a reinforcement of the tight junctions in the gut barrier [32,33]. In fact, Rescigno et al. used a similar co-culture system and described that DCs open the tight junctions between epithelial cells, send dendrites outside the epithelium and directly sample bacteria. In addition, because DCs express tight-junction proteins, such as occludin, claudin 1 and zonula occludens 1, the integrity of the epithelial barrier is preserved [22]. As for IL-6 upregulation, certain probiotic strains are known to be potent inducers of IL-6. Thus, Weiss et al. have described the capacity of 27 species of lactobacilli and 16 species of bifidobacteria to induce the expression of IL-6 and other cytokine mRNAs in murine bone marrow-derived DCs [34]. Given the fact that IL-6 is clearly implicated in inflammatory bowel disease and other chronic diseases and cancer, its serum levels are considered markers of inflammatory diseases and anti-IL-6 treatments are under investigation [35].

TLR4 recognizes LPS present in the wall of Gram-negative bacteria and plays a key role in the defense against pathogens. This TLR was also upregulated in the co-cultures treated with *B. infantis* IM-1^®^, similar to the results described by Bermúdez-Brito et al. in DC cultures treated with *L. paracasei* CNCM I-4034 [23]. These results suggest that probiotics modulate cytokine production in immune cells through TLR stimulation.

Altogether, our results point to the clear protective and anti-inflammatory effects of *B. infantis* IM-1^®^ and its supernatant on intestinal epithelial and DCs in vitro. We are aware that a limitation of our study is that these results obtained in vitro should be validated in the future with in vivo and human studies.

## 4. Materials and Methods

### 4.1. Probiotic Strain

The probiotic strain, *Bifidobacterium longum* subsp. *infantis* CECT 7210, has been characterized and described previously [13].

### 4.2. Preparation of Bacteria and Cell-Free Culture Supernatant

*B. longum* subsp. *infantis* CECT 7210 was routinely anaerobically cultured for 24 h at 37 °C in de Man–Rogosa–Sharpe (MRS) broth medium (Oxoid, Basingstoke, UK), supplemented with 0.05% (wt/vol) cysteine (Sigma-Aldrich, St. Louis, MO, USA). The supernatant of the culture was collected by centrifugation at 12,000× *g* for 10 min, neutralized to pH 7.0 by the addition of 1 N NaOH, passed through a 0.22-μm pore size filter unit (Minisart hydrophilic syringe filter; Sartorius Stedim Biotech GmbH, Goettingen, Germany) and stored at −20 °C until use. The supernatant was added to the cell culture medium at a concentration of 7% vol/vol.

*Escherichia coli* CECT 515 was provided by the Spanish Type Culture Collection (CECT; Burjasot, Valencia, Spain) and aerobically cultured for 24 h at 37 °C in tryptone soy broth (Panreac Química, Barcelona, Spain). The bacterial concentration in the suspension was estimated from the optical density at 600 nm and adjusted to the appropriate concentration by dilution with PBS. *E. coli* CECT 515 was used because it is a strain of recognized pathogenic action; it produces diarrhea through the secretion of toxins [36,37]. This strain is equivalent to WDCM 0090 and NCTC 9001.

### 4.3. Cells

Two cell types were used in this work: Caco-2, an epithelial cell line obtained from colorectal adenocarcinoma (ATCC, Manassas, VA, USA); and human dendritic cells (DCs) from peripheral blood, purchased from Lonza (reference K3CC-2701; Basel, Switzerland).

Caco-2 cells were cultured in Dulbecco’s modified Eagle’s medium (DMEM) (Sigma-Aldrich) supplemented with 10% inactivated FBS, 1% glutamine, penicillin G (0.1 U/mL) and streptomycin (0.1 mg/mL). The cells were cultured at 37 °C in an atmosphere of 5% CO_2_ and 95% air. Caco-2 cells were stimulated with *E. coli* CECT 515 for 4 or 8 h. LPS at 60 ng/mL (Sigma-Aldrich) was used as a positive control. We used 1 Caco-2 cell per 100 *E. coli* cells and 1 Caco-2 cell per 30 probiotic cells.

### 4.4. Co-Cultures of Caco-2 and DCs

The co-culture of Caco-2 and DCs described by Bermúdez-Brito et al. (2015) [23] was used. Caco-2 cells were seeded in the upper chamber of a transwell filter (3 μm pore size, 6.5 mm diameter; Corning, Corning, NY, USA), grown to confluence and kept post confluence for 15 days until a trans-epithelial resistance of 1000 Ω cm^2^ was reached. After this culture time, Caco-2 cells were fully differentiated into enterocytes. Transwell inserts containing adherent monolayers of Caco-2 cells were then positioned upside down and 5 × 10^4^ DCs were placed on the membrane. The DCs were allowed to adhere to the filters at 37 °C. After 4 h, the transwell inserts were turned right side up and placed in 24-well plates containing 1–1.5 × 10^5^ DCs/well in the lower chamber.

*B. longum* subsp. *infantis* CECT 7210, its supernatant or *E. coli* CECT 515 were added from the apical surface (top chamber), and the cells were incubated for 1 h. The culture medium was then replaced by the maintenance medium LGM3 (Lonza, reference CC-3211). Co-cultures were then incubated for 23 additional hours. After incubation, the culture supernatants were collected from the lower chamber for cytokine analysis. LPS at 60 ng/mL (Sigma-Aldrich) was used as a positive control. Negative controls were unstimulated co-cultures.

### 4.5. Cytokine Quantification in Culture Supernatants

Cytokine production was measured by immunoassay with the MILLIplex^TM^ kit (Linco Research Inc., St. Charles, MO, USA), using the Luminex 200 system according to the manufacturer’s instructions. IL-1β, IL-4, IL-8, IL-10, IL-12 (p70), TNF-α, IFN-γ TGF-β1, TGF-β2 and TGF-β3 were analyzed.

### 4.6. RNA Isolation from Cell Lysates and qRT-PCR

The total RNA was isolated from cells using the RNeasy Mini Kit (Qiagen, Barcelona, Spain), following the manufacturer’s recommendations. Complementary DNA (cDNA) was synthesized using the iScript advanced cDNA Synthesis Kit (Bio-Rad Laboratories, Hercules, CA, USA). cDNA was amplified with SYBR Green PCR Master Mix (Applied Biosystems, Glasgow, UK) and an ABI Prism 7900 instrument (Applied Biosystems, Foster City, CA, USA).

The specific primers used were: TLR4, assay ID: qHsaCED0037607; IL-6, Assay ID: qHsaCED0044677; and occludin, assay ID: qHsaCED0038290 (Bio-Rad, Coralville, IA, USA). q-PCR data were normalized to the β-actin gene (assay ID: qHsaCED0036269). The PCR conditions were 1 cycle of 95 °C for 10 min followed by 40 cycles of 95 °C for 15 s and 60 °C for 1 min. The 2^−∆∆Ct^ method was used for relative quantification. Changes in gene expression were expressed as fold change versus unstimulated Caco-2 cells.

### 4.7. Flow Cytometry

For viability analysis, Caco-2 cells were detached with Trypsin-EDTA (0.25%) at 37 °C and centrifuged at 300× *g* for 5 min. The supernatant was discarded, and the pellet was resuspended in PBS at 10^6^ cells/mL and then centrifuged at 300× *g* for 5 min. The cells were permeabilized with cold ethanol (78%) for 5 min, washed in PBS, and labeled with fluorescein diacetate (1 mg/mL) and propidium iodide (0.01 mg/mL) (Sigma-Aldrich, St. Louis, MO, USA) for 15 min. Cell viability was then analyzed on a FACSCanto II cytometer (BD, Palo Alto, CA, USA) equipped with three laser lines: 405 nm, 488 nm and 633 nm. Fluorescence measurements were captured on FL3 (fluorescein isothiocyanate, FITC) and FL4 (phycoerythrin, PE) detectors equipped with two BP filters (530 ± 30 nm and 570 ± 20 nm). FACSDiva (San Jose, CA, USA software (BD), version 7.2, was used as a data analysis tool.

### 4.8. Statistical Analysis

All results are expressed as the mean ± standard error of the mean (SEM) of three independent experiments. Multiple comparisons of normally distributed variables were analyzed by a one-way analysis of variance (ANOVA) followed by a Tukey post hoc test. For non-normally distributed variables multiple comparisons were performed using the Kruskal–Wallis ANOVA and a Dunn’s post hoc test. Analyses were performed using GraphPad Prism version 8.0.1 for Windows (GraphPad Software, San Diego, CA, USA). *p* < 0.05 was considered statistically significant.

## Figures and Tables

**Figure 1 ijms-23-10813-f001:**
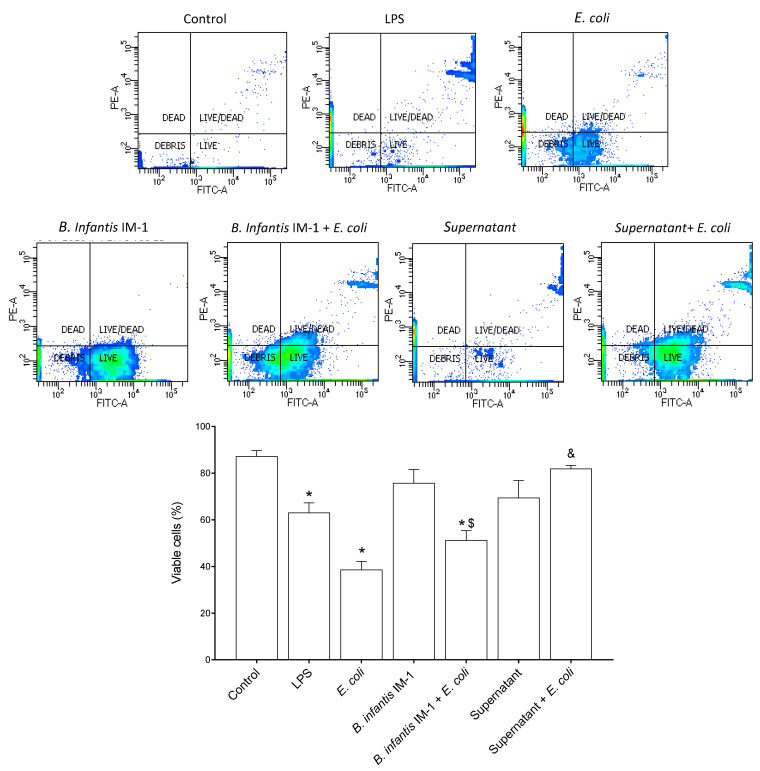
Flow cytometry analysis of Caco-2 cell viability. Viability of Caco-2 cells treated for 8 h with either LPS, *E. coli* (1:100 Caco-2: *E. coli*), *B. infantis* IM-1^®^ (1:30 Caco-2: *B. infantis* IM-1^®^) or *B. infantis* IM-1^®^ supernatant in the presence or absence of *E. coli*. Results are expressed in percentages as the mean ± SEM of three independent experiments. * *p* < 0.05 vs. control (untreated cells); ^$^
*p* < 0.05 vs. *B. infantis* IM-1^®^; ^&^
*p* < 0.01 vs. *B. infantis* IM-1^®^ + *E. coli*.

**Figure 2 ijms-23-10813-f002:**
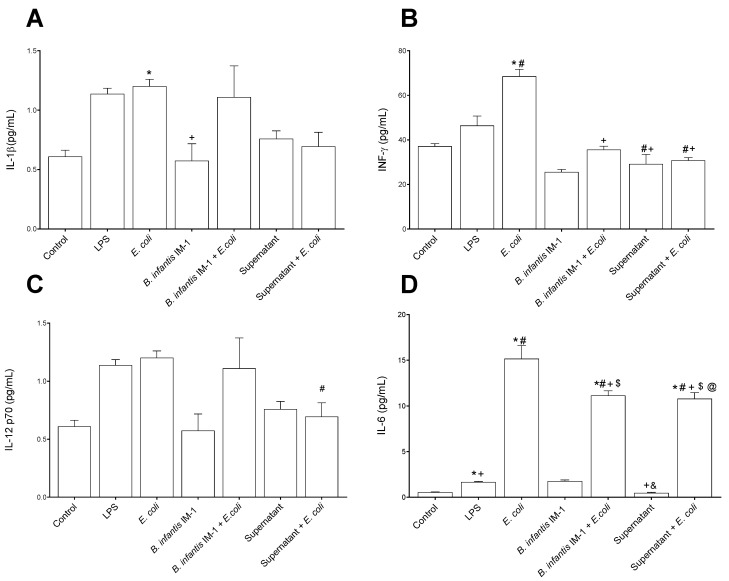
Cytokine levels in the culture medium of Caco-2 cells after 8 h of treatments. (**A**) IL-1β. (**B**) IFN-γ. (**C**) IL-12 p70. (**D**) IL-6. (**E**) IL-8. (**F**) TNFα. (**G**) IL-10. (**H**) IL-4. Results are expressed in pg/mL as the mean ± SEM of three independent experiments. * *p* < 0.05 vs. control (untreated cells); ^#^
*p* < 0.05 vs. LPS; ^+^
*p* < 0.05 vs. *E. coli*; ^$^
*p* < 0.01 vs. *B. infantis* IM-1^®^; ^&^
*p* < 0.01 vs. *B. infantis* IM-1^®^ + *E. coli;* ^@^
*p* < 0.001 vs. supernatant.

**Figure 3 ijms-23-10813-f003:**
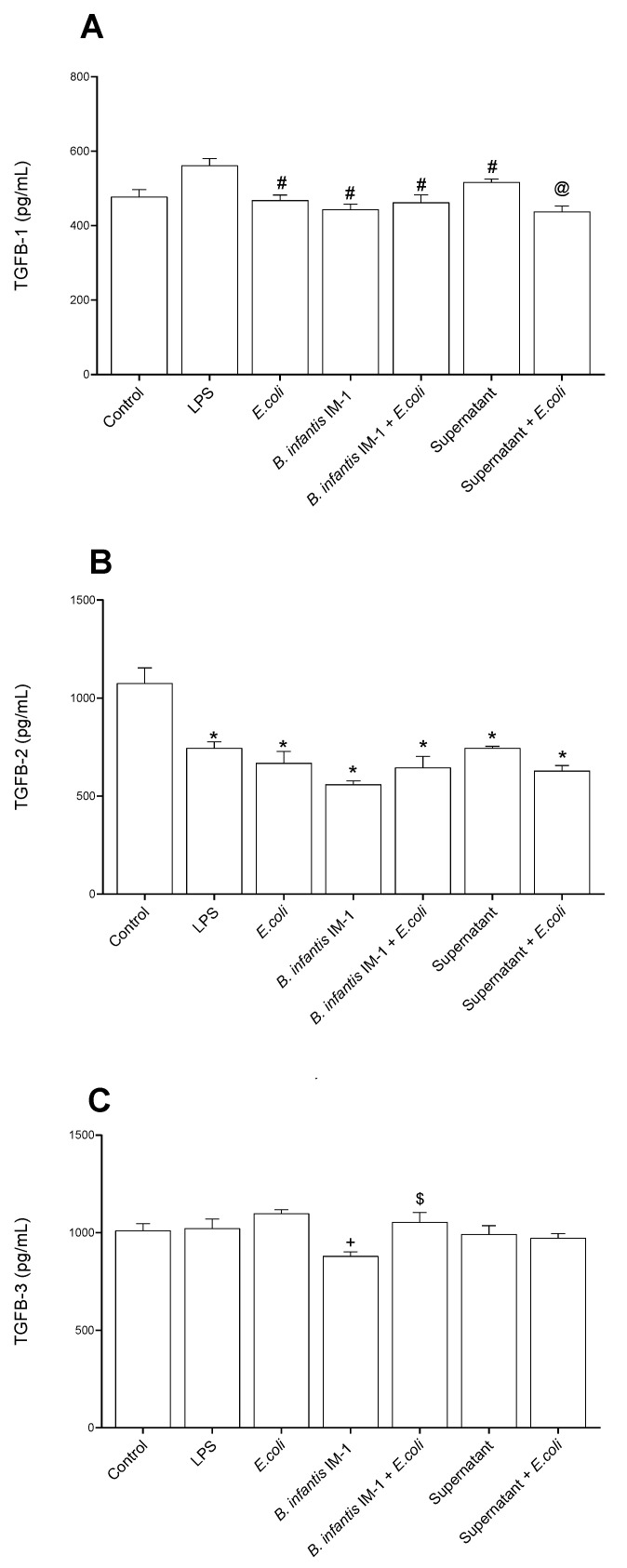
TGF-β levels in the culture medium of Caco-2 cells after 8 h of treatments. (**A**) TGF-β1. (**B**) TGF-β2. (**C**) TGF-β3. Results are expressed in pg/mL as the mean ± SEM of three independent experiments. * *p* < 0.05 vs. control (untreated cells); ^#^
*p* < 0.05 vs. LPS; ^+^
*p* < 0.05 vs. *E. coli;* ^$^
*p* < 0.01 vs. *B. infantis* IM-1^®^; ^@^
*p* < 0.001 vs. supernatant.

**Figure 4 ijms-23-10813-f004:**
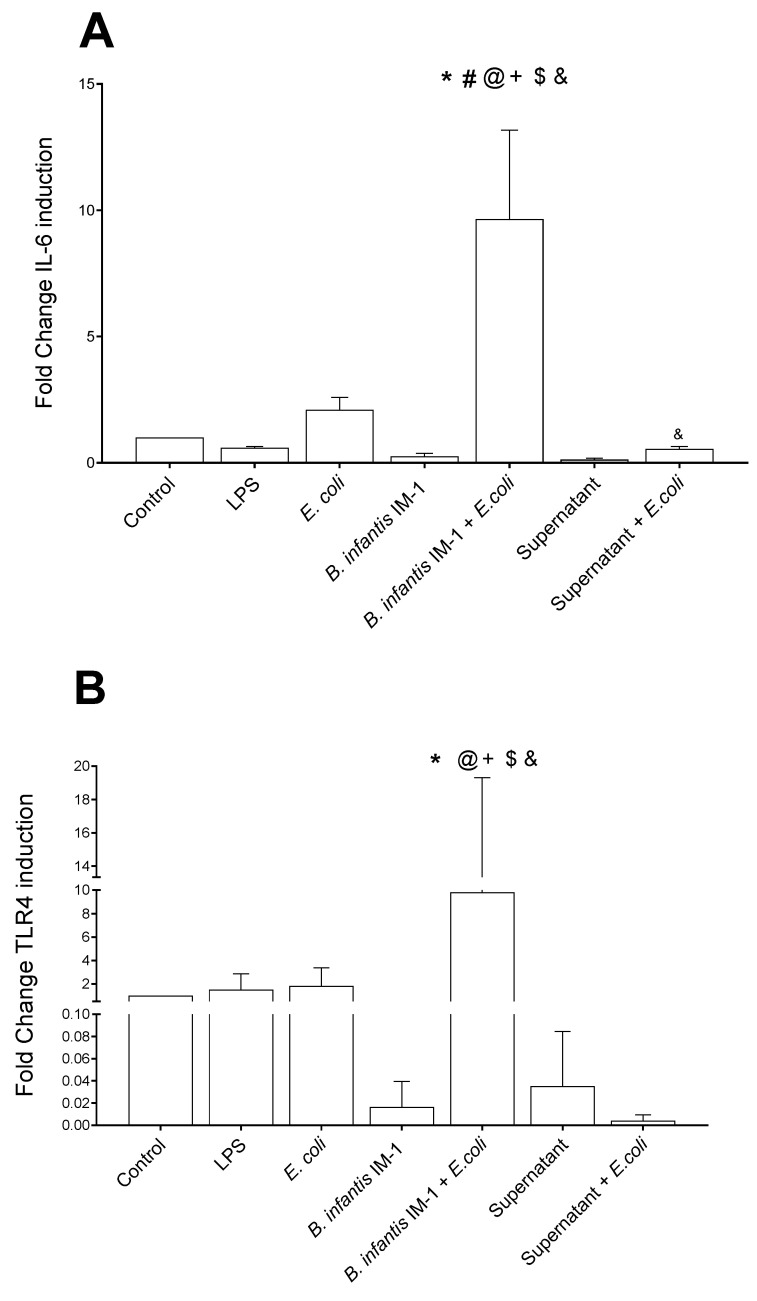
Expression of IL-6 (**A**), TLR-4 (**B**) and occludin (**C**) genes by Caco-2 cells determined by qRT-PCR. Results are expressed as the mean ± SEM of three independent experiments. Changes in gene expression are expressed as fold change versus unstimulated Caco-2 cells. * *p* < 0.05 vs. control (untreated cells); ^#^
*p* < 0.05 vs. LPS; ^+^
*p* < 0.05 vs. *E. coli;* ^$^
*p* < 0.01 vs. *B. infantis* IM-1^®^; ^&^
*p* < 0.01 vs. *B. infantis* IM-1^®^ + *E.coli*; ^@^
*p* < 0.001 vs. supernatant.

**Figure 5 ijms-23-10813-f005:**
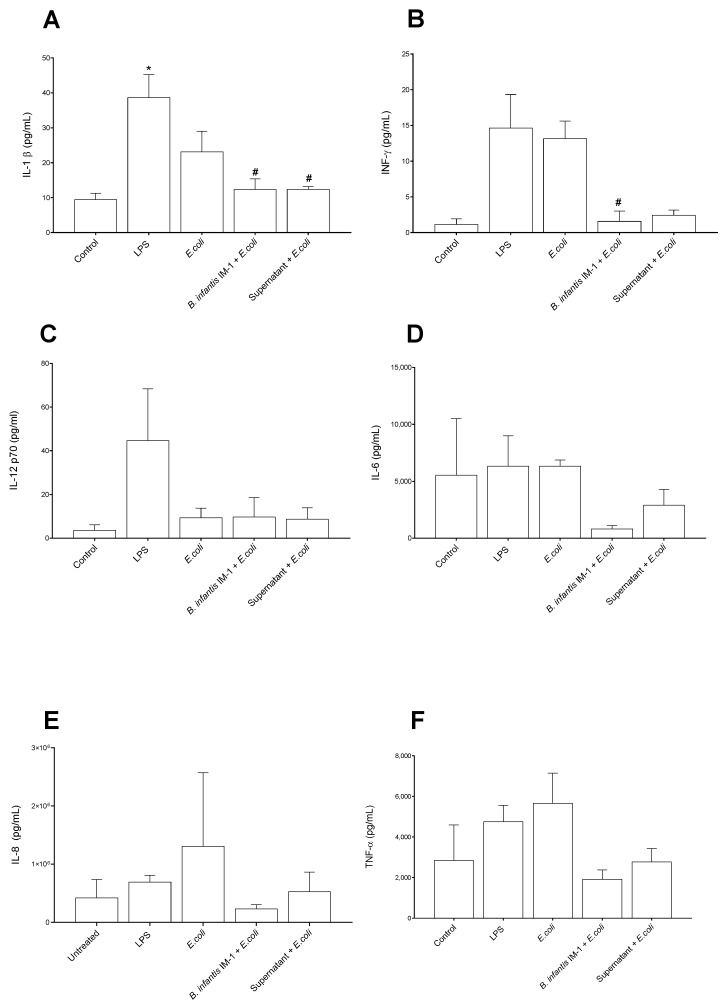
Cytokine levels in the culture medium of co-cultures of Caco-2 and DCs after 1 h of treatments. (**A**) IL-1β. (**B**) IFN-γ. (**C**) IL-12 p70. (**D**) IL-6. (**E**) IL-8. (**F**) TNFα. (**G**) IL-10. (**H**) IL-4. Results are expressed in pg/mL as the mean ± SEM of three independent experiments. * *p* < 0.05 vs. control (untreated cells); ^#^
*p* < 0.05 vs. LPS.

**Figure 6 ijms-23-10813-f006:**
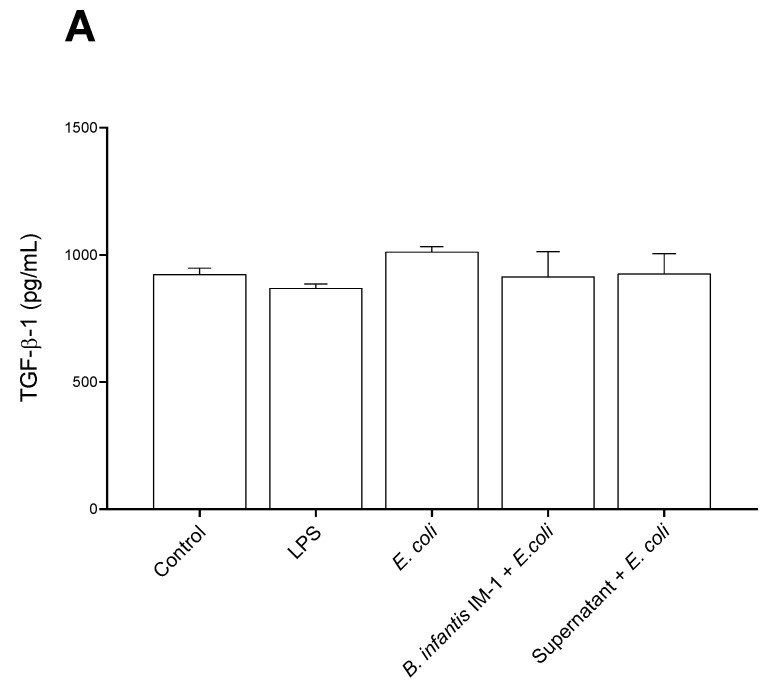
TGF-β levels in the culture medium of co-cultures of Caco-2 and DCs after 1 h of treatments. (**A**) TGF-β1. (**B**) TGF-β2. (**C**) TGF-β3. Results are expressed in pg/mL as the mean ± SEM of three independent experiments.

## Data Availability

The datasets generated during and/or analyzed during the current study are available from the corresponding author upon reasonable request.

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
