# Peer review of "Bifidobacterium longum subsp. infantis CECT 7210 Reduces Inflammatory Cytokine Secretion in Caco-2 Cells Cultured in the Presence of Escherichia coli CECT 515"

_ijms, 2022, doi:10.3390/ijms231810813_

Round 1
Reviewer 1 Report
1- Overuse of citation number 1 in the introduction, it is a revision of one of the authors, please use original articles for each of the affirmations.
2-Some species names are not in italics (see paragraphs 64-69).
3-Why did the authors use Escherichia coli strain CECT 515, this information should appear in material and methods.
4-Figure 1C. In the figure caption it indicates "Viability of Caco-2 cells treated for 8 hours with either LPS, E. coli (1:100 Caco-2:E. coli), B. infantis 102 IM-1® (1:30 Caco-2: B. infantis IM-1®) or B. infantis IM-1® supernatant in the presence or absence 103 of E. coli.". However, in the material and methods it indicates that the treatment was 1 hour. The incubation time is not clear. In addition, how does pH variation influence the viability of Caco2?. This point should be further reviewed.
5-Paragraph 244; the whole paragraph is in italics.
6-Figure 3 is placed before the text, please change it.
7-Figure 4C. The control bar has no upper limit.
8-Figures 5 and 6. What is the reason for hiding the bars of B. infantis IM-1 and supernatant? Please add them. It makes no sense and is misleading that the title of the section is "B. infantis IM-1® and its supernatant reduce the pro-inflammatory response by co-cultures of Caco-2 and DCs" when there is precisely no data for these samples.
9-Line 166, the strain code is missing.
10-Line 213 "Up-regulation of occludin points to a reinforcement of the tight junctions in the gut barrier.". Missing citation.
11-Line 218-220. Expand on the information about the effect of IL-6 and the importance of decreasing its production.
12-It would be useful for the authors to discuss the non-increase of the anti-inflammatory cytokine IL-10.
Author Response
The authors would like to thank the reviewer for his/her respected comments and effort made during the review process, which are highly appreciated.
1- Overuse of citation number 1 in the introduction, it is a revision of one of the authors, please use original articles for each of the affirmations. As, requested, we have added new references in this paragraph to support the statements. The following references have been added:
Liu Y, Wang J, Wu C. Modulation of Gut Microbiota and Immune System by Probiotics, Pre-biotics, and Post-biotics. Front Nutr. 2022 Jan 3;8:634897. doi: 10.3389/fnut.2021.634897. PMID: 35047537; PMCID: PMC8761849.
Reference # 10
Suda K, Matsuda K. How Microbes Affect Depression: Underlying Mechanisms via the Gut-Brain Axis and the Modulating Role of Probiotics. Int J Mol Sci. 2022 Jan 21;23(3):1172. doi: 10.3390/ijms23031172. PMID: 35163104; PMCID: PMC8835211.
Reference # 11
Zeuthen LH, Fink LN, Frøkiaer H. Toll-like receptor 2 and nucleotide-binding oligomerization domain-2 play divergent roles in the recognition of gut-derived lactobacilli and bifidobacteria in dendritic cells. Immunology. 2008 Aug;124(4):489-502. doi: 10.1111/j.1365-2567.2007.02800.x. Epub 2008 Jan 24. PMID: 18217947; PMCID: PMC2492941.
Reference # 12
2-Some species names are not in italics (see paragraphs 64-69). We have revised all species throughout the manuscript and corrected those that were not in italics (lines 67-69)
3-Why did the authors use Escherichia coli strain CECT 515, this information should appear in material and methods. We decided to use E. coli CECT 515 because it is a strain of recognized pathogenic action (it produces diarrhea through the secretion of toxins) and it is deposited in the Spanish Type Culture Collection (that is what CECT stands for). We have added this information to the methodology section, as suggested (lines 260-266).
4-Figure 1C. In the figure caption it indicates "Viability of Caco-2 cells treated for 8 hours with either LPS, E. coli (1:100 Caco-2:E. coli), B. infantis 102 IM-1® (1:30 Caco-2: B. infantis IM-1®) or B. infantis IM-1® supernatant in the presence or absence 103 of E. coli.". However, in the material and methods it indicates that the treatment was 1 hour. The incubation time is not clear. In addition, how does pH variation influence the viability of Caco2? This point should be further reviewed. Incubation times were different for Caco-2 cells alone and co-cultures: 8 h for single cells and 1 h for co-cultures. Incubation time for co-cultures was 1 h based on reference 23. As for the pH, it was neutralized to 7,4 before adding it to the cells. Therefore, there is no variation in pH.
5-Paragraph 244; the whole paragraph is in italics. This paragraph has been correctly formatted.
6-Figure 3 is placed before the text, please change it. This figure has been moved after the text, as requested by the reviewer.
7-Figure 4C. The control bar has no upper limit. This error has been fixed and now the control bar appears entirely.
8-Figures 5 and 6. What is the reason for hiding the bars of B. infantis IM-1 and supernatant? Please add them. It makes no sense and is misleading that the title of the section is "B. infantis IM-1® and its supernatant reduce the pro-inflammatory response by co-cultures of Caco-2 and DCs" when there is precisely no data for these samples. In these 2 experiments, cells were not treated with B. infantis or the supernatant alone because we wished to study the effect in the presence of E. coli. We believe that the title is still valid/true.
9-Line 166, the strain code is missing. The code has been added (line 169)
10-Line 213 "Up-regulation of occludin points to a reinforcement of the tight junctions in the gut barrier.". Missing citation. Two references have been added to support this statement:
Braniste V, Leveque M, Buisson-Brenac C, Bueno L, Fioramonti J, Houdeau E. Oestradiol decreases colonic permeability through oestrogen receptor beta-mediated up-regulation of occludin and junctional adhesion molecule-A in epithelial cells. J Physiol. 2009 Jul 1;587(Pt 13):3317-28. doi: 10.1113/jphysiol.2009.169300.
Reference # 32
Dokladny K, Ye D, Kennedy JC, Moseley PL, Ma TY. Cellular and molecular mechanisms of heat stress-induced up-regulation of occludin protein expression: regulatory role of heat shock factor-1. Am J Pathol. 2008 Mar;172(3):659-70. doi: 10.2353/ajpath.2008.070522.
Reference # 33
11-Line 218-220. Expand on the information about the effect of IL-6 and the importance of decreasing its production. This issue has been extended, as requested: “Given the fact that IL-6 is clearly implicated in inflammatory bowel disease and other chronic diseases and cancer, its serum levels are considered markers of inflammatory diseases and anti-IL-6 treatments are under investigation [35]” (lines 234-237).
12-It would be useful for the authors to discuss the non-increase of the anti-inflammatory cytokine IL-10. We have included a new paragraph stating that:
“As for anti-inflammatory cytokines we focused on IL-10 and IL-4. B. infantis IM-1® did not affect their secretion, but the results were consistent in single cultures of Caco-2 cells and co-cultures. Our IL-4 and IL-10 results coincide with those obtained by different authors that used other probiotic strains [25,26] However, production of other anti-inflammatory cytokines by B. infantis IM-1® should not be ruled out. (lines 191-195).

Reviewer 2 Report
The aim of the study was to investigate the anti-inflammatory potential of the well-known and characterized probiotic strain B. longum subsp. infantis CECT 7210 in vitro using co-culture of epithelial and dendritic cells. Although the strain effectively reduces episodes of diarrhea in piglets and humans, the characterization of its immunomodulatory activity could expand its possible use in medicine. Even though the manuscript is well structured and flows well, I have several comments.
1. The main weakness of the manuscript is that the authors did not analyse the composition of the probiotic supernatant/conditioned medium. Information on active components (SCFAs, other metabolites or bacteriocins, etc.) should improve the quality of the manuscript. Since the strain is well characterized, I suggest supplementing the discussion with at least speculations about the active substances that could be anti-inflammatory.
2. Figure 1 A,B regarding the selection of optimal dose of bacteria and treatment time should be included as supplementary material.
3. Lines 45-56: Instead of a single cited review (Ref n. 1), add more specific references.
4. Lines 65-67: Format bacteria´s names in italics (Cronobacter, Salmonella, …)
5. Fig. 4 C – part of the graph is missing in the control column.
6. Lines 166-177: This part is repeated and was already mentioned in the introduction (lines 63-80).
7. Lines 244-248 : Use normal font, not italics.
Author Response
The authors would like to thank the reviewer for his/her respected comments and effort made during the review process, which are highly appreciated.
The aim of the study was to investigate the anti-inflammatory potential of the well-known and characterized probiotic strain B. longum subsp. infantis CECT 7210 in vitro using co-culture of epithelial and dendritic cells. Although the strain effectively reduces episodes of diarrhea in piglets and humans, the characterization of its immunomodulatory activity could expand its possible use in medicine. Even though the manuscript is well structured and flows well, I have several comments.
- The main weakness of the manuscript is that the authors did not analyse the composition of the probiotic supernatant/conditioned medium. Information on active components (SCFAs, other metabolites or bacteriocins, etc.) should improve the quality of the manuscript. Since the strain is well characterized, I suggest supplementing the discussion with at least speculations about the active substances that could be anti-inflammatory. We have included the following explanation to the discussion (lines 218-223) along with new references: “B. infantis IM-1® is known to secrete short chain fatty acids (SCFA) to the culture medium. Studies have shown a higher production of butyric, lactic, acetic and valeric acids in the intestine of animals fed this probiotic strain. Although we did not analyze the composition of the cell-free supernatant, we speculate that the anti-inflammatory effects observed in this study were due to its SCFA content. Other components of the supernatant might also be responsible or contribute to these effects”.
- Figure 1 A, B regarding the selection of optimal dose of bacteria and treatment time should be included as supplementary material. We would rather keep the figures as they are to better follow the train of thought.
- Lines 45-56: Instead of a single cited review (Ref n. 1), add more specific references. As, requested, we have added new references in this paragraph to support the statements:
Liu Y, Wang J, Wu C. Modulation of Gut Microbiota and Immune System by Probiotics, Pre-biotics, and Post-biotics. Front Nutr. 2022 Jan 3;8:634897. doi: 10.3389/fnut.2021.634897. PMID: 35047537; PMCID: PMC8761849.
Reference # 10
Suda K, Matsuda K. How Microbes Affect Depression: Underlying Mechanisms via the Gut-Brain Axis and the Modulating Role of Probiotics. Int J Mol Sci. 2022 Jan 21;23(3):1172. doi: 10.3390/ijms23031172. PMID: 35163104; PMCID: PMC8835211.
Reference # 11
Zeuthen LH, Fink LN, Frøkiaer H. Toll-like receptor 2 and nucleotide-binding oligomerization domain-2 play divergent roles in the recognition of gut-derived lactobacilli and bifidobacteria in dendritic cells. Immunology. 2008 Aug;124(4):489-502. doi: 10.1111/j.1365-2567.2007.02800.x. Epub 2008 Jan 24. PMID: 18217947; PMCID: PMC2492941.
Reference # 12
- Lines 65-67: Format bacteria´s names in italics (Cronobacter, Salmonella, …). We have revised all bacterial names and corrected those that were not in italics (lines 67-69)
- Fig. 4 C – part of the graph is missing in the control column. This error has been fixed.
- Lines 166-177: This part is repeated and was already mentioned in the introduction (lines 63-80). The first sentence of the discussion (three lines) is actually a quick summary of a couple of paragraphs in the introduction. We do not see any problem in repeating information in a different manner. If the reviewer prefers the sentence to be deleted we will comply, but we would rather keep the sentence so the reader can follow our train of thought.
- Lines 244-248 : Use normal font, not italics. We have formatted this paragraph correctly.

Round 2
Reviewer 1 Report
no more suggestions
Author Response
The authors would like to thank the reviewer for her/ his efforts during the review process.